# Outpatient Antibiotic Prescriptions in France: Patients and Providers Characteristics and Impact of the COVID-19 Pandemic

**DOI:** 10.3390/antibiotics11050643

**Published:** 2022-05-11

**Authors:** Wilfried BARA, Christian Brun-Buisson, Bruno Coignard, Laurence Watier

**Affiliations:** 1Center for Research in Epidemiology and Population Health (CESP), INSERM U1018, Université Paris-Saclay, UVSQ, 78180 Montigny-Le-Bretonneux, France; wilfried.bara@santepubliquefrance.fr (W.B.); chris.brunb@gmail.com (C.B.-B.); 2Institut Pasteur, Epidemiology and Modelling of Antibiotic Evasion (EMAE), Université Paris-Cité, 75015 Paris, France; 3Santé publique France, the French National Public Health Agency, 94415 Saint-Maurice, France; bruno.coignard@santepubliquefrance.fr; 4Université Paris Est Créteil, 94010 Créteil, France

**Keywords:** antibiotics, outpatient prescription, primary care physicians, COVID-19, ambulatory care, French National Health Insurance Databases (SNDS), healthcare professionals

## Abstract

In France, despite several successive plans to control antimicrobial resistance, antibiotic use remains high in the outpatient setting. This study aims to better understand outpatient antibiotic use and prescription in order to identify tailored targets for future public health actions. Using data from the French National Health Data System, we described and compared the individual characteristics of patients with and without an antibiotic prescription. The prescribed antibiotics (ATC-J01) were detailed and compared between 2019 and 2020. Antibiotic prescribing indicators that take prescriber activity into account were estimated and compared. Patients who were female, advanced age, and the presence of comorbidities were associated with antibiotic prescriptions. The overall prescription rate was estimated at 134 per 1000 consultations and 326 per 1000 patients seen in 2019. General practitioners (GPs), dentists and paediatricians were associated with 78.0%, 12.2% and 2.2% of antibiotic prescriptions, respectively, with high prescription rates (391, 447, and 313 p. 1000 patients seen, respectively). In comparison with 2019, this rate decreased in 2020 for paediatricians (−30.4%) and GPs (−17.9%) whereas it increased among dentists (+17.9%). The reduction was twice as high among the male prescribers than among their female counterparts (−26.6 and −12.0, respectively). The reduction in prescriptions observed in 2020 (−18.2%) was more marked in children (−35.8%) but less so among individuals ≥65 years (−13.1%) and those with comorbidities (−12.5%). The decrease in penicillin prescriptions represents 67.3% of the overall reduction observed in 2020. The heterogeneous decrease in prescriptions by age and antibiotic class could be explained by the impact of COVID-19 control measures on the spread of respiratory viruses; thus, a substantial proportion of the prescriptions avoided in 2020 is likely inappropriate, particularly among children. In order to keep the rate of prescriptions comparable to that observed in 2020, male prescribers, paediatricians and GPs should be encouraged to maintain that level, while a campaign to raise awareness of the appropriate use of antibiotics should be aimed at dentists in particular.

## 1. Introduction

Excessive antibiotic use has largely contributed to the emergence and global spread of increasingly difficult-to-treat resistant bacteria [1]. In 2019, the annual number of deaths due to antibiotic-resistant infections was estimated at over 35,000 in the USA [2]. In France, the latest estimates assess the impact of bacterial resistance at approximately 5500 deaths in 2015 [3] and 139,000 infections in 2016 [4]. Antibiotic use, whether justified or not, is one of the main factors contributing to the spread of bacterial resistance, particularly through the selective pressure it places on pathogenic or commensal bacteria [5,6,7]. In France, despite three successive multisectoral action plans to control antibiotic resistance, antibiotic use remains high and has changed very little since 2010. According to the European Centre for Disease Prevention and Control (ECDC), France was the fourth largest consumer of antibiotics in the outpatient setting in 2019 with 23.3 DDD per 1000 inhabitants per day, 30% higher than the European average [8].

Studies have shown that over 50% of outpatient prescriptions are issued inappropriately [2], mostly for acute respiratory infections primarily of viral origin [9,10]. However, controlling the volume of antibiotics dispensed requires an understanding of the factors that could influence their prescription. Many European and US studies published between 2014 and 2020 have attempted to identify variables related to outpatient antibiotic prescriptions [10,11,12,13,14,15,16,17] but were often limited to viral respiratory infections or to a fraction of the population [16,17,18]. In Europe (Greece, Sweden, Switzerland, UK, Spain) several analyses using medical-administrative databases have shown that individual patient characteristics were associated with antibiotics prescriptions [13,14,19,20,21,22]. In addition, several studies reported differing providers practices in the US [23], Canada [24], or Asia [25], suggesting that physicians’ preferences influenced prescribing decisions independently of the patient’s condition. Furthermore, the significant reduction in prescriptions observed during the COVID-19 pandemic in several countries including France [26,27,28,29] suggests that there is room for progress in reducing the volume of antibiotics dispensed in the outpatient sector in the context of a full care offering.

Since 2018, the monitoring of antibiotic use in France has been carried out under the auspices of the French National Public Health Agency (Santé Publique France). This monitoring is used to define awareness-raising strategies aimed at the public and healthcare professionals (HCPs). For sound, evidence-based planning, it is important to identify new levers for action in order to better target the relevant populations. To our knowledge, no studies have examined the individual factors associated with antibiotic prescription in France. The primary objective of this study is to describe the epidemiology of physicians’ prescriptions of antibiotics in the outpatient setting and identify the associated individual characteristics of the patients and prescribers. Finally, the impact of the COVID-19 pandemic and associated control measures on outpatient antibiotic prescriptions was analysed.

## 2. Results

In 2019, more than 58 million patients received at least one reimbursement for outpatient medical care for a total of almost 50 million antibiotic prescription reimbursements. Over 15% of these patients (and the same percentage of antibiotic prescriptions) were excluded because they were not affiliated to the general scheme (GS), were not residing in mainland France, or died during the year (Figure 1). The exclusion of HCPs who were inactive (no consultations, aged ≥75 years), non-medical, and those whose annual consultations number appeared to be outliers, and outpatient services associated with a missing or unselected HCP, led to the additional exclusion of 7.7% of patients corresponding to 18% of antibiotic prescription reimbursements, of which more than 10% were hospital prescriptions. The final study sample included approximately 45 million patients, for 136,380 HCPs and 34 million antibiotic prescription reimbursements.

### 2.1. Patients

Of the selected patients, 41.4% were reimbursed for at least one antibiotic prescription (With-Atb) in 2019. The majority were female (58.9%), of a working age between 15–65 years (60.4%) and residing in urban areas (68.6%). In terms of health status, 20.4% had at least one long-term illnesses (LTI), mostly chronic respiratory diseases, cardiovascular diseases, diabetes and cancer. Among those aged under 60, 13.2% were recipients of complementary universal health insurance (CMUc). In terms of healthcare system use, 63.6% of the patients With-Atb had at least five visits and 42.6% received at least two antibiotic prescriptions throughout the year. Compared to patients Without-Atb, they were more often female, aged under 5 or over 65 years, more likely to reside in the northern region of Hauts-de-France and less likely to reside in the southern Auvergne-Rhône-Alpes (Table 1). They were more likely to have comorbidities and twice as likely to suffer from chronic respiratory diseases and to have more HCP visits (median: 6 vs. 3) than patients Without-Atb. Patients With-Atb were comparable to patients Without-Atb in terms of area of residence and socio-economic conditions (FDep). These trends persisted in a multivariate analysis (data not shown).

In 2020, 34.7% of the analyzed population received at least one antibiotic prescription (Table 1). Their socio-demographic characteristics were similar to those from 2019 (Table 1). However, the number of patients exposed to antibiotics (With-Atb) decreased by 13.5% compared to 2019. The decrease was most marked among those under 15 years of age (−25.1%), or with more than two prescriptions per year (−25.8%), but less in those aged over 64 (−9.9%). Among those under 15 years of age, the reduction mainly affected recipients of multiple prescriptions (Appendix A).

### 2.2. Prescribers

In 2019, 136,380 (68.8%) of the 198,135 prescribers identified were selected. They prescribed 81.9% of reimbursed antibiotic prescriptions to the selected patients (Figure 1). The majority of prescribers were male, with an average age of 52 years and 19.4 years of experience (Table 2). Most were in private practice and affiliated to sector 1. They were primarily general practitioners (GPs) (42.2%), other medical subspecialty (27.2%), dentists (26.6%), dermatologists (2.1%), or paediatricians (1.9%). GPs accounted for 76.6% of outpatient visits and issued 78% of antibiotic prescriptions, followed by dentists, other subspecialists, paediatricians, and dermatologists. As expected, in 2020 the number of selected HCPs (Appendix A) and thus their characteristics were almost the same (Appendix A).

### 2.3. Antibiotic Prescriptions

In 2019, 34 million outpatient antibiotic prescriptions were issued in the selected population. The majority of these prescriptions were for female patients or those aged 15–65 (Table 3). Approximately two thirds of the prescriptions were issued by male prescribers, or by those over 50 years of age. The most commonly prescribed therapeutic classes were penicillins (52.8%), macrolides (15.0%), third-generation cephalosporins (3GCs, 8.5%), other antibacterials such as fosfomycin (6.4%), quinolones (5.6%) and tetracyclines (3.3%). Except for dermatologists, for whom tetracyclines accounted for 82.9% of their prescriptions, HCPs primarily prescribed penicillins (Figure 2, Appendix A). The second most prescribed class of antibiotics was macrolides for GPs, combinations of antibacterials (spiramycin and metronidazole) for dentists, and 3GCs for paediatricians.

In 2020, less than 28 million outpatient antibiotic prescriptions were issued (Table 3 and Appendix A), corresponding to a significant reduction (−18.2%) in antibiotic prescriptions relative to 2019. The most marked reductions in proportions were for the other beta-lactam groups, followed by penicillins; however, the reduction in the latter represents 67.3% of the overall reduction observed between 2019 and 2020. The reduction was most marked in patients aged less than 15, where J01C antibiotics accounted for over 70% of antibiotic use (Appendix A). Despite the overall reduction observed, a few molecules increased compared to 2019, particularly azithromycin (+10.1%). GPs and paediatricians experienced the greatest reductions in the number of prescriptions, decreasing by 21.8% and 34.4%, respectively (Table 3).

### 2.4. Prescribing Indicators

The overall prescription rate was 754 per 1000 patients in 2019, compared to 597 per 1000 in 2020 (Figure 3a). This rate was higher among women and almost twice as high among those under 5 years of age. After 5 years, it increased regularly with the age of the patients. Among prescribers, the overall prescription rate was estimated at 134 per 1000 consultations (Table 2) and 325 per 1000 patients seen in 2019 (Figure 3b). By subspecialty, the prescription rate per 1000 patients seen was 448, 389, 312 and 117 for dentists, GPs, paediatricians and dermatologists, respectively. Female prescribers had a lower rate than their male counterparts in 2019.

In 2020, despite an overall reduction in consultations (9.4%) and antibiotic prescriptions (18.2%) compared to 2019, the change in indicators was heterogeneous between specialties (Figure 3b). Among GPs and paediatricians, the number of prescriptions per 1000 patients seen decreased by 17.9% and 30.4%, respectively. Conversely, this rate increased by 17.9% among dentists. Among male prescribers, the observed decrease was twice as high as in female prescribers, resulting in an identical prescribing rate (256 p. 1000) between both genders in 2020.

## 3. Discussion

This study provides an overview of outpatient antibiotic prescription in France, an essential step for understanding outpatient prescription practices, specifically the individual characteristics of the patients and prescribers associated with these prescriptions. Our analyses show that antibiotic prescription recipients are more often female, belong to extreme age groups (<5 years or ≥65 years) and often have comorbidities, being twice as likely to suffer from chronic respiratory diseases. Among those under the age of 60, antibiotic prescription recipients are more often vulnerable (CMUc). These results are consistent with the literature [13,30,31,32]. Comparisons between 2019 and during the COVID-19 pandemic show that nearly 20% of antibiotic prescriptions could be avoided, particularly in the younger age group.

GPs and dentists are associated with over 90% of outpatient antibiotic prescriptions. In proportion, GPs are the primary prescribers followed by dentists and other medical specialties (78.0%, 12.2%, and 5.9%, respectively). The prescription rate for dentists was 447 per 1000 patients, while that of GPs was 391 per 1000. Relative to their activity, dentists are thus the highest prescribers of antibiotics in the primary care sector, issuing an antibiotics prescription to almost one in two patients on average. A recent systematic review reports that the majority of dentists routinely prescribe antibiotics for patients who require dental implant surgery [33]. However, the proposed treatment regimens (preoperative and/or post-operative) do not always follow the recommendations and may indicate an over-prescription of antibiotics. We also observed that male prescribers over 50 years of age are associated with a high rate of antibiotic prescriptions. Similar results were found in a study in the United States [11]. A French survey reports that having graduated more recently is associated with a more rational use of antibiotics [34], which could be explained by a better understanding of the issues of antibiotic resistance and greater awareness regarding the appropriate use of antibiotics in the training provided to the youngest generation.

Furthermore, despite the overall reduction in prescriptions in 2020 for all medical subspecialties, dentists appear to have maintained or even increased their level of prescriptions. Indeed, prescriptions of amoxicillin alone (J01CA) and in combinations (J01CR) by dentists increased by 3.8% and 8.2%, respectively, compared to 2019 (Appendix A). Similar results were observed in an Australian study that reported a small reduction in antibiotic prescriptions issued by dentists in 2020 compared to 2019, although prescriptions for other drugs by dentists decreased significantly [35]. Some authors have reported that dentists tend to issue prescriptions for broader-spectrum antibiotics, such as amoxicillin in combination with clavulanic acid, when no surgical treatment is considered [35,36]. Limited access to primary care including dental offices during the first lockdown due to the SARS-CoV-2 pandemic (17 March to 10 May 2020) could explain a more frequent use of antibiotic therapy as a result of delayed surgical treatment. However, the use of broad-spectrum antibiotics by dentists remains a concern as they are particularly prone to fostering bacterial resistance. These results suggest that dentists are a major target and have a key role to play in national strategies to combat antibiotic resistance.

In 2019, the annual prescription rate was 754 per 1000 patients and 134 per 1000 consultations. The most commonly prescribed antibiotics were penicillins, macrolides and other beta-lactams (primarily 3GCs). A liberal prescription of broad-spectrum antibiotics was also observed in most European countries [37] and is likely due to the frequent implementation of probabilistic treatments in the outpatient sector. However, the prescription rate observed in France (754 per 1000 patients) appears higher than that observed in other high-income countries. In 2015, it was 626 per 1000 patients in England and 323 per 1000 inhabitants in Sweden [31,37].

We estimated the reduction in the number of antibiotic prescriptions and of the number of patients exposed to these prescriptions at −18.2% and −13.5% in 2020 as compared with 2019, respectively. Similar results were observed in Europe where, in terms of prescription, reductions of 24% and 13.5% were observed in Spain and the United Kingdom, respectively [28,38]. The reduction in antibiotic prescriptions was especially apparent in children and primarily concerned antibiotics commonly used to treat respiratory infections (amoxicillin alone or in combination with enzyme inhibitors). The reduction in these antibiotics accounts for more than two thirds of the overall reduction observed in the general population and almost three quarters (73.5) for those aged 0–15 years. The overall reduction in antibiotic prescription can be explained, in part, by decreased use of care following the restrictions (lockdown, curfew, etc.) related to the SARS-CoV-2 pandemic. However, the heterogeneous reduction in prescriptions by age group and according to therapeutic classes could be explained by the promotion of preventive measures (closure of schools and nurseries, mask wearing, physical distancing, hygiene practices, etc.) which have strongly contributed to limiting not only the spread of COVID-19, but also of other respiratory infectious agents [39,40]. Indeed, during the winter of 2020–2021, a dramatic reduction in the circulation of winter viruses in France and therefore consultations for episodes of acute respiratory infection was observed [41,42,43]. As respiratory infections are predominantly of viral origin and affect children in particular, a significant proportion of the prescriptions avoided, especially among children, could therefore be unnecessary or inappropriate, as has been suggested by other authors in the literature [42,44,45]. The COVID-19 pandemic thus offers an opportunity to identify levers for action for improved antibiotic use and to combat antibiotic resistance in general.

This study is based on data from the SNDS, which include all outpatient care subject to reimbursement for the entire population. Thus, due to the high statistical power related to the large number of subjects, we did not use statistical tests to compare proportions or ratio. In fact, very marginal differences were statistically significant and a multivariate analysis did not provide additional information. Nevertheless, this database has some limitations, in particular the absence of reliable indicators of precariousness, which meant we were unable to conduct an in-depth study of the effect of social inequalities on antibiotic use. In addition, this study was restricted to recipients affiliated to the main public health insurance scheme and strictly to outpatient antibiotic prescriptions, because the specialty of HCPs is not recorded for the vast majority of hospital-based outpatient’s medical services. Indeed, hospital prescriptions of antibiotics are not negligible and certainly affect a specific population. Finally, the reason for the prescription or duration of treatment are not available. Therefore, it is not possible to discuss the clinical relevance of the prescriptions or whether the antibiotics dispensed are appropriate for the conditions treated or not. However, this analysis remains important as a first step in identifying the priority targets (patients/prescribers) to be explored and towards whom awareness-raising efforts should be directed.

## 4. Materials and Methods

### 4.1. Source of Data

The SNDS is a medical-administrative data warehouse fuelled by a prospective and almost exhaustive collection of all the services reimbursed by the national health insurance (Assurance Maladie). It essentially contains the individual data used for the billing and reimbursement of outpatient care services and public and private hospital data collected in the medicalisation of the information systems programme PMSI by the Agence technique de l’information sur l’hospitalisation [the technical agency for information on hospital care—ATIH] [46]. It also contains a repository that links health professionals (HCPs) to each of the services provided.

By means of its permanent access to data from the SNDS, Santé Publique France has the authorisation from the Commission Nationale Informatique et Libertés [French Data Protection Authority—CNIL] requested to conduct this study.

### 4.2. Study Populations

Patients who received at least one reimbursement for outpatient medical services in 2019 and in 2020 were identified. Patients not affiliated to the main public health insurance scheme [general scheme—GS], were not residing in metropolitan France and who died during the year of interest were excluded. This selection covered approximately 76% of the French population in 2015 [46].

The prescribing physicians selected were GPs, other medical subspecialists or dentists. They were required to be in actual practice, under 75 years of age and to have a reasonable annual number of consultations (<9600 consultations/year). Midwives were excluded due to the specificity of their patients and prescriptions. HCPs unrecorded in the database were also excluded (these were primarily prescribers associated with hospital-based medical services).

Among all medical services received by the selected patients, those associated with an unselected or unknown HCP were excluded. As a result, patients who did not receive all of their outpatient medical services from a selected prescriber were also excluded.

The selected patients were then classified into two groups: those without reimbursement for antibiotic prescriptions (Without-Atb) and those with at least one reimbursement for antibiotic prescriptions (With-Atb). Only antibiotics for systemic use (Anatomical Therapeutic Chemical classification—ATC—J01) were considered.

### 4.3. Variables Studied

The following socio-demographic characteristics of the patients were considered: sex, age (<5; 5–15; 15–45; 45–65 and ≥65 years), region of residence (n = 13), area of residence (rural, urban), complementary universal health insurance (CMUc: yes/no among those aged <60 years) and social deprivation index of the town of residence estimated in 2015 (quintiles: Q1—least disadvantaged to Q5—most disadvantaged) [47] (see Appendix A). The health status of patients was approximated by the number of long-term illnesses (LTIs) eligible for 100% reimbursement of healthcare for the disease concerned, and the comorbidities most often encountered. These were identified using algorithms developed by the French national health insurance fund [48]. Patients use of the ambulatory health system was approximated by the number of consultations (1–2; 3–4; 5–9 and ≥10), as well as the number of antibiotic prescriptions (1; 2; 3; and ≥4) among those with at least one antibiotic reimbursement.

The prescribers were characterised by sex, age (<45; 45–60; ≥60), subspecialty (GP, Dental Surgeon, Paediatrician, Dermatologist, Other subspecialty), years of experience (<15; 15–30; ≥30), type of practice (private, mixed, salaried), and conventional status within the healthcare system (sector 1, sector 2) (see Appendix A). The annual number of patients seen, and of consultations and prescriptions, was used to assess their activity. As a patient may consult several prescribers, the total number of patients seen for all prescribers during the year exceeded the size of the French population.

Antibiotics were categorised into eight levels according to the level 3 ATC classification: tetracyclines (J01A); penicillins (J01C); other beta-lactam antibacterials (J01D); sulphonamides and trimethoprim (J01E); macrolides (J01F); quinolones (J01M); combinations of antibacterials (J01R) and other antibacterial drugs (J01X). Additionally, penicillins with extended-spectrum (J01CA), combinations of penicillins (J01CR), third-generation cephalosporins (J01DD) and azithromycin (J01FA10) were singled out.

### 4.4. Prescribing Indicators

Three prescribing indicators were developed. The number of prescriptions was standardized by: (1) total number of patients selected (With-Atb and Without-Atb), (2) number of consultations, and (3) number of patients seen. These indicators were estimated overall and for selected characteristics of patients and prescribers. For the dentists, the number of patients seen was preferred to the number of consultations. Indeed, the concept of consultation alone does not reflect the activity of dentists because they carry out far more routine dental procedures than consultations.

### 4.5. Statistical Analyses

For each year studied, the socio-demographic characteristics of patients and antibiotic prescribers are described. Qualitative variables are reported as frequencies (percentages) and quantitative variables as mean and standard deviation (SD) or median and interquartile range (IQR). The characteristics of patients With-Atb were compared to those Without-Atb. However, due to the size of the population, no statistical tests were performed and the different rates are reported without their confidence interval [49]. Antibiotic prescriptions are described according to the socio-demographic characteristics of the patients, providers and therapeutic groups, and differences between 2019 and 2020 calculated. The prescribing indicators were estimated annually for all patients and prescribers, and then stratified by age, sex and medical subspecialty.

All analyses were performed using the SAS GUIDE software (version 7.13, SAS Institute Inc., Cary, NC, USA).

## 5. Conclusions

The recipients of antibiotic prescriptions in France are more often female, children or with complementary universal health insurance. At the provider level, GPs, dentists, male prescribers and those over 50 years of age were associated with a higher rate of antibiotic prescription. Regarding COVID-19 impact, a relatively large decrease in prescribing among children than the rest of the population was observed from 2019 to 2020. The COVID-19 pandemic helped to achieve the objectives of reducing antibiotic prescribing in the community in an unexpected way. However, additional studies are needed to better understand drivers of such a reduction and sustained antimicrobial stewardship actions, such as the ones planned in the new French national strategy for preventing antimicrobial resistance [50], will be needed in order for this decrease in antimicrobial prescriptions to be sustainable.

## Figures and Tables

**Figure 1 antibiotics-11-00643-f001:**
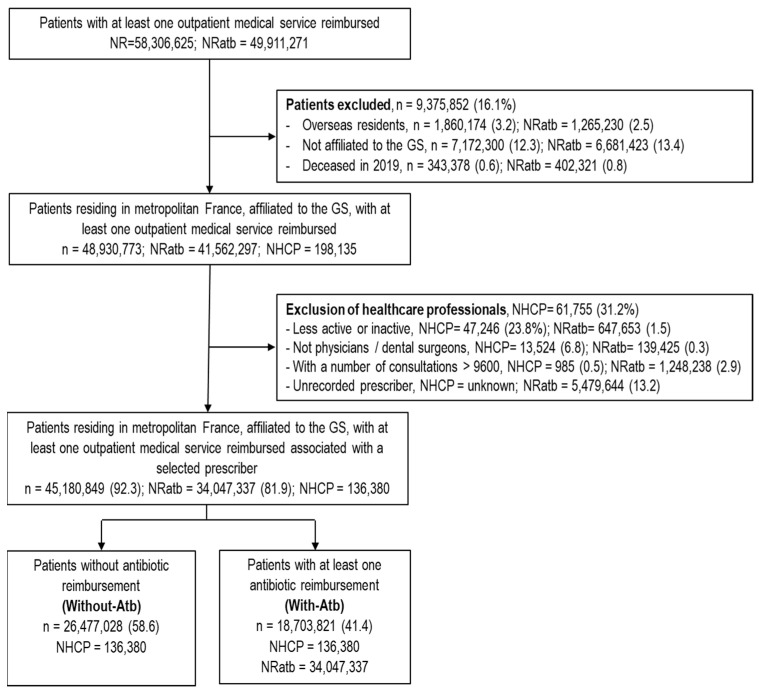
Flowchart of patient and prescriber selection from French health insurance databases (SNDS)—2019. n: Number of patients, NRatb: Number of reimbursements for antibiotics, NHCP: Number of healthcare professionals, HCP: Healthcare professional, GS: General scheme (main public health insurance scheme in France).

**Figure 2 antibiotics-11-00643-f002:**
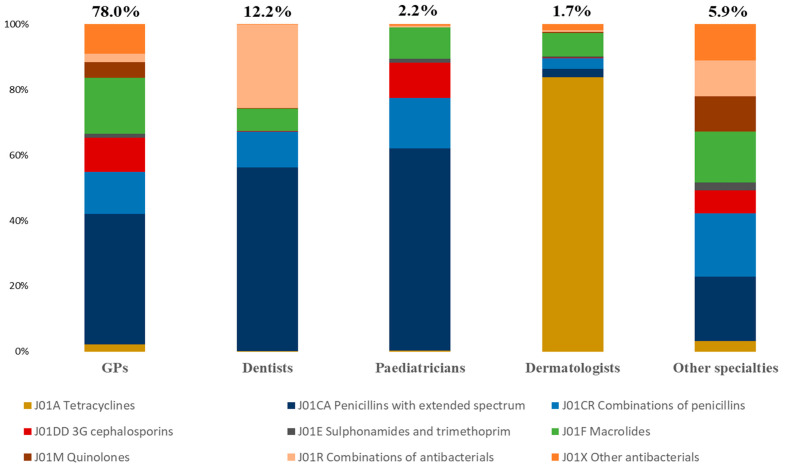
Distribution of therapeutic classes of antibiotics prescribed according to subspecialty of prescriber, 2019, SNDS.

**Figure 3 antibiotics-11-00643-f003:**
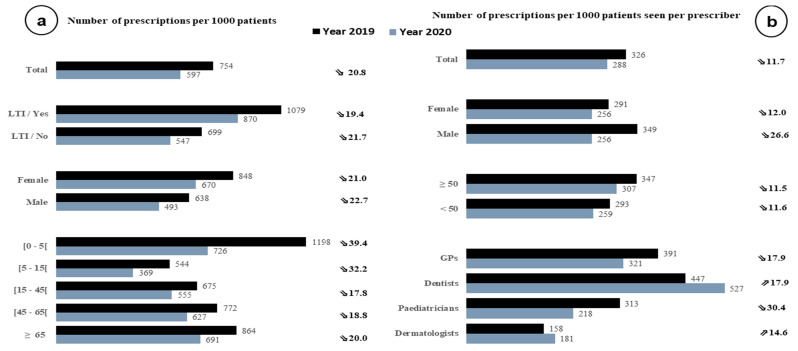
Antibiotic prescription rate (**a**) according to age, sex, presence of comorbidities (long-term illness, LTI) among patients; (**b**) according to age, sex, and subspecialty of prescribers—2019 and 2020, SNDS. To the right of each figure is the percentage (%) change in the indicators in 2020 compared to 2019.

**Table 1 antibiotics-11-00643-t001:** Socio-demographic characteristics of patients with (With-Atb) or without (Without-Atb) at least one antibiotic prescription reimbursement, 2019 and 2020, SNDS.

	Year 2019				Year 2020			
Characteristic	Without-Atb		With-Atb		Without-Atb		With-Atb	
	*n*	%	*n*	%	*n*	%	*n*	%
	26,477,028	58.6	18,703,821	41.4	30,492,755	65.3	16,174,425	34.7
Sex								
Male	12,949,452	48.9	7,681,528	41.1	14,804,256	48.6	6,616,141	41.0
Female	13,527,448	51.1	11,022,193	58.9	15,688,489	51.4	9,532,590	59.0
Age in years								
[0–4]	1,243,498	4.7	1,528,973	8.2	1,616,231	5.3	1,133,255	7.0
[5–14]	3,751,652	14.2	1,972,825	10.6	4,170,206	13.7	1,491,278	9.2
[15–44]	9,787,330	37.0	6,482,426	34.7	10,787,768	35.4	5,695,477	35.2
[45–64]	6,632,635	25.1	4,806,033	25.7	7,746,405	25.4	4,327,717	26.8
≥65	5,061,799	19.1	3,913,564	20.9	6,172,145	20.2	3,526,698	21.8
Median (IQR ^1^)	40 (19–59)	-	42 (20–61)	-	41 (20–60)	-	43 (23–62)	-
Number of long-term illnesses (LTI)								
0	21,927,912	84.1	14,596,193	79.6	24,868,480	82.8	12,422,238	78.4
1	3,199,233	12.3	2,734,653	14.9	3,981,177	13.3	2,514,039	15.9
≥2	959,540	3.7	1,011,325	5.5	1,175,989	3.9	903,731	5.7
Comorbidities								
Chronic respiratory	979,191	4.0	1,465,344	8.2	1,282,053	4.5	1,301,885	8.4
Cardiovascular	1,556,421	6.3	1,418,797	8.0	1,970,846	6.9	1,307,048	8.4
Diabetes	1,347,042	5.5	1,149,997	6.5	1,663,939	5.9	1,059,982	6.8
Types of cancer	972,843	4.0	948,449	5.3	121,226	4.3	871,066	5.6
Complementary universal health insurance CMUc								
Population < 60 years	19,909,570	75.2	13,642,472	73.0	22,984,453	74.1	11,616,489	71.9
Yes	2,026,487	10.2	1,801,283	13.2	2,712,922	11.8	1,782,625	15.3
Residence area								
Rural	8,233,538	32.0	5,735,200	31.4	9,549,105	31.7	4,927,830	31.3
Urban	17,496,612	68.0	12,507,620	68.6	20,574,085	68.3	10,819,973	68.7
Regions of residence								
Auvergne-Rhône-Alpes	3,512,752	13.6	2,139,269	11.7	3,922,996	13.3	1,802,396	11.4
Bourgogne-Franche-Comté	1,071,646	4.2	792.034	4.3	1,220,548	4.1	672.196	4.3
Brittany	1,407,558	5.5	878.389	4.8	1,603,757	5.4	746.349	4.7
Centre-Val de Loire	1,043,472	4.0	667.860	3.6	1,161,773	3.9	568.459	3.6
Corsica	96.200	0.4	95.367	0.5	120.946	0.4	87.748	0.6
Grand Est	2,224,183	8.6	163.7714	8.9	2,517,511	8.5	1,389,502	8.8
Hauts-de-France	2,274,070	8.8	1,903,928	10.4	2,635,567	8.9	1,645,956	10.4
Île-de-France	4,707,424	18.2	3,289,329	17.9	5,367,918	18.1	2,800,595	17.8
Normandy	1,246,605	4.8	914.042	5.0	1,437,588	4.8	781.745	5.0
Nouvelle-Aquitaine	2,295,033	8.9	1,733,969	9.5	2,705,874	9.1	1,523,883	9.7
Occitanie	2,318,158	9.0	1,747,434	9.5	2,728,435	9.2	1,518,379	9.6
Pays de la Loire	1,623,283	6.3	910.753	5.0	1,820,213	6.2	778.280	4.9
Provence-Alpes-Côte d’Azur	2,001,770	7.8	1,631,496	8.9	2,372,981	8.0	1,457,778	9.2
Social deprivation index								
Q1 (least disadvantaged)	5,332,126	20.6	3,668,352	20.0	6,277,265	20.8	3,149,281	19.9
Q2	5,357,867	20.7	3,700,019	20.2	6,256,429	20.7	3,174,668	20.1
Q3	5,282,014	20.5	3,732,332	20.4	6,152,617	20.4	3,212,878	20.3
Q4	5,072,874	19.6	3,584,209	19.6	5,892,691	19.5	3,100,271	19.6
Q5 (most disadvantaged)	4,785,247	18.5	3,625,804	19.8	5,660,298	18.7	3,164,046	20.0
Number of consultations								
[1–2]	9,961,149	37.6	2,851,634	15.3	13,492,844	44.2	3,504,330	21.7
[3–4]	7,109,098	26.8	3,969,442	21.2	8,119,038	26.6	3,889,795	24.1
[5–9]	6,980,363	26.4	7,272,078	38.9	6,770,662	22.2	5,684,991	35.1
≥10	2,426,304	9.2	4,610,567	24.7	2,110,211	6.9	3,095,309	19.1
Number of prescriptions per patient								
1	-		10,734,648	57.4	-		9,892,615	61.2
2	-		4,352,182	23.3	-		3,598,891	22.3
3	-		1,878,977	10.1	-		1,433,011	8.9
4 or more	-		1,737,914	9.3	-		1,249,908	7.7

^1^ InterQuartile Range.

**Table 2 antibiotics-11-00643-t002:** Socio-demographic characteristics of antibiotic prescribers, overall and according to subspecialty, 2019, SNDS.

	All HCPs ^1^		GPs ^2^		Dentists		Paediatricians		Dermatologists		Other	
	*n*	%	*n*	%	*n*	%	*n*	%	*n*	%	*n*	%
Sex												
Women	54,471	40.1	23,359	40.7	15,543	43.2	1717	65.9	1937	69.5	11,915	32.3
Men	81,201	59.9	34,086	59.3	20,416	56.8	888	34.1	852	30.5	24,959	67.7
Age, in years												
<50	52,580	39.1	21,071	37.0	18,595	51.9	888	34.8	641	23.4	11,385	31.3
≥50	81,900	60.9	35,927	63.0	17,226	48.1	1664	65.2	2096	76.6	24,959	67.7
Mean ± SD ^3^	52 ± 12.2	-	52.7 ± 11.8		47.5 ± 12.9		54.2 ± 11.2		56.4 ± 10.1		54.8 ± 10.8	
Years of experience												
Junior ≤ 15 years	52,164	39.0	21,360	37.8	14,393	40.5	1191	46.9	710	25.9	14,510	39.7
Intermediate 15–30 years	49,399	36.9	21,000	37.2	12,318	34.7	836	32.9	1175	42.9	14,070	38.5
Senior ≥ 30 years	32,236	24.1	14,090	25.0	8838	24.9	514	20.2	8838	24.9	7938	21.7
Mean ± SD	19.4 ± 12.5	-	19.7 ± 12.7		19.2 ± 12.6		17.4 ± 12.7		22.7 ± 11.5		18.9 ± 12.0	
Type of practice												
Private	127,288	93.4	54,496	94.7	35,023	96.4	2239	85.8	2488	88.9	33,042	89.3
Mixed	8854	6.5	2924	5.1	1289	3.6	370	14.2	311	11.1	3960	10.7
Salaried	146	0.1	143	0.3	-		-		-		2	0.0
Conventional status												
Sector 1	110,028	81.2	53,315	93.5	36,225	100.0	1502	57.7	1552	55.9	17,434	47.3
Sector 2	25,469	18.8	3714	6.5	-		1100	42.3	1225	44.1	19,430	52.7
Activity												
Consultations	253,421,882	-	194,404,033	76.6	10,351,166	4.1	6,363,102	2.5	4,636,225	1.8	37,667,356	14.9
Patients seen	104,286,371	-	67,863,794	65.1	9,306,050	8.9	2,370,138	2.3	3,558,126	3.4	21,188,263	20.3
Prescriptions ^4^	34,047,337	-	26,566,106	78.0	4,164,192	12.2	742,006	2.2	563,016	1.7	2,012,017	5.9
Prescription per 1000 consultations	134	-	137	-	-	-	117	-	121	-	53	
All Providers	136,380	-	57,573	42.2	36,330	26.6	2612	1.9	2800	2.1	37,065	27.2

^1^ Healthcare professionals; ^2^ General practitioners; ^3^ Standard Deviation; ^4^ Antibiotic prescription.

**Table 3 antibiotics-11-00643-t003:** Distribution of prescriptions according to some socio-demographic characteristics of recipients and prescribers and therapeutic classes of antibiotics, 2019 and 2020, SNDS.

	Number of Antibiotic Prescriptions per Year				∆ 2019–2020, *n* (%)
	2019	%	2020	%	
**Socio-demographic characteristics of recipients**					
**Sex**					
Male	13,186,680	38.7	10,714,622	38.5	**↓** 2,472,058 (18.7)
Female	20,860,657	61.3	17,137,640	61.5	**↓** 3,723,017 (17.8)
**Long-term illness (LTI)**					
No	25,516,739	74.9	20,392,046	73.2	**↓** 5,124,693 (20.1)
Yes	8,530,598	25.1	7,460,216	26.8	**↓** 1,070,382 (12.5)
**Age, in years**					
[0–4]	3,325,334	9.8	2,023,230	7.3	**↓** 1,302,104 (39.2)
[5–14]	3,121,348	9.2	2,118,555	7.6	**↓** 1,002,793 (32.1)
[15–44]	10,996,596	32.3	9,306,849	33.4	**↓** 1,689,747 (15.4)
[45–64]	8,837,130	26.0	7,653,068	27.5	**↓** 1,184,062 (13.4)
≥65	7,766,929	22.8	6,750,560	24.2	**↓** 1,016,369 (13.1)
**Socio-demographic characteristics of prescribers**					
**Sex**					
Male	22,205,010	65.4	17,967,750	64.7	**↓** 4,237,260 (19.1)
Female	11,760,626	34.6	9,808,765	35.3	**↓** 1,951,861 (16.6)
**Age, in years**					
<50	11,691,815	34.6	9,972,842	36.0	**↓** 1,718,973 (14.7)
≥50	22,085,071	65.4	17,699,356	64.0	**↓** 4,385,715 (19.9)
**Years of experience**					
Junior ≤15 years	11,597,693	34.4	9,930,308	35.9	**↓** 1,667,385 (14.4)
Intermediate 15–30 years	13,886,717	41.1	10,771,737	39.0	**↓** 3,114,980 (22.4)
Senior ≥ 30 years	8,310,943	24.6	6,925,732	25.1	**↓** 1,385,211 (16.7)
**Provider speciality**					
General practitioners (GPs)	26,566,106	78.0	20,786,542	74.6	**↓** 5,779,564 (21.8)
Dental surgeons	4,164,192	12.2	4,208,530	15.1	**↑** 44,338 (1.1)
Paediatricians	742,006	2.2	486,840	1.8	**↓** 255,166 (34.4)
Dermatologists	563,016	1.7	560,958	2.0	**↓** 2058 (0.4)
Other specialisms	2,012,017	5.9	1,809,392	6.5	**↓** 202,625 (10.1)
**Therapeutic class**					
**J01A Tetracyclines**	1,132,043	3.3	1,126,738	4.1	**↓** 5305 (0.5)
**J01C Penicillins**	17,991,482	52.8	13,820,626	49.6	**↓** 4,170,856 (23.2)
J01CA Penicillins with extended spectrum	13,472,862	39.6	9,897,223	35.5	**↓** 3,575,639 (26.5)
J01CR Combinations of penicillins	4,306,958	12.7	3,731,584	13.4	**↓** 575,374 (13.3)
**J01D Other beta-lactams**	3,425,607	10.1	2,377,932	8.5	**↓** 1,047,675 (30.6)
J01DD 3G cephalosporins	2,904,226	8.5	2,098,008	7.5	**↓** 806,218 (27.8)
**J01E Sulphonamides and trimethoprim**	375,785	1.1	387,661	1.4	**↑** 11,876 (3.2)
**J01F Macrolides**	5,110,174	15.0	4,371,751	15.7	**↓** 738,423 (14.4)
J01FA10: Azithromycin	1,637,132	4.8	1,801,862	6.5	**↑** 164,730 (10.1)
**J01M Quinolones**	1,508,069	5.6	1,339,817	4.8	**↓** 168,252 (11.1)
**J01R Combinations of antibacterials ^1^**	1,934,692	5.7	1,909,135	6.9	**↓** 25,557 (1.3)
**J01X Other antibacterials**	2,550,694	7.5	2,505,787	9.0	**↓** 44,907 (1.8)
**All prescriptions**	34,047,337		27,852,262		**↓** 6,195,075 (18.2)

^1^ Spiramycin and metronidazole.

## Data Availability

Data cannot be shared publicly because access to French administrative medical databases is regulated by the French Data Protection Authority—CNIL. Temporary access for studies and research is available on request from the National Institute for Health Data (INDS).

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
