# Peer review of "Outpatient Antibiotic Prescriptions in France: Patients and Providers Characteristics and Impact of the COVID-19 Pandemic"

_antibiotics, 2022, doi:10.3390/antibiotics11050643_

Round 1

Reviewer 1 Report

  1. Line 11: Change ‘antibioresistance’ to antimicrobial (or antibiotic) resistance
  2. Keywords: What is SNDS?
  3. Line 57: A ‘.’ is missing after the references
  4. Results: Delete the word count after the word ‘results’
  5. Figure 1: Part of the figure (on the right side) is not shown. Please correct it
  6. Line 92: LTI? Please define abbreviations when first used
  7. Line 93: CMUc? Please define abbreviations when first used
  8. Line 100: number in parentheses are %? If yes, add the symbol (%)
  9. Line 118: GPs. Even though it is obvious, it should be defined when first used
  10. Table 2: Could add in the footnote the definitions of all abbreviations, such as HCPs, GPs
  11. Lines 130 and 131: if the numbers in the parentheses are percentages, add the corresponding symbol (%)
  12. Line 134: if the numbers in the parentheses are percentages, add the corresponding symbol (%)
  13. Table 3: Please correct the brackets in the left column
  14. Table 3: Specialisms? Maybe you mean specialists
  15. Table 3: Add GPs to the footnotes with a definition
  16. Figure 3: Something is wrong with the legend (grey is not defined) and the numbers on the right side of the figure. Probably some information is lost when fitting on the page?
  17. Materials and methods: Delete the word count
  18. Was there an ethics approval or waiver for this study? If yes, it should be stated in the Methods section and at the end of the manuscript before the references
  19. Supplementary material: Please define all abbreviations in the footnotes of the tables
  20. A conclusion section summarizing the main findings could be added after the methods section
  21. The authors could add a small comment regarding how their data could aid in the reduction of unnecessary antimicrobial prescribing, also suggesting methods to implement antimicrobial stewardship practices in the community setting

Author Response

Line 11: Change ‘antibioresistance’ to antimicrobial (or antibiotic) resistance

This has been modified.

Keywords: What is SNDS?

This has been modified by French National health insurance databases (SNDS)

Line 57: A ‘.’ is missing after the references

Indeed, it has been added

Results: Delete the word count after the word ‘results’

Word count has been removed

Figure 1: Part of the figure (on the right side) is not shown. Please correct it

Done

Line 92: LTI? Please define abbreviations when first used

Indeed, we changed the order of appearance of the material/methods and results paragraphs in the last moments before submission. Therefore, we missed the definition of some abbreviations when they were first used. They are all defined now.

Line 93: CMUc? Please define abbreviations when first used

Done! Please find attached its meaning that is indicated in the appendices

Complementary universal health insurance (CMUc): it is used as an indicator of precariousness since its allocation is based on individual resources. However, this indicator cannot be used after the age of 60 since the amount of the minimum old-age pension paid to persons with low resources exceeds the limit for receiving CMUc.

Line 100: number in parentheses are %? If yes, add the symbol (%)

Added

Line 118: GPs. Even though it is obvious, it should be defined when first used

Defined

Table 2: Could add in the footnote the definitions of all abbreviations, such as HCPs, GPs

Done

Lines 130 and 131: if the numbers in the parentheses are percentages, add the corresponding symbol (%)

Added

Line 134: if the numbers in the parentheses are percentages, add the corresponding symbol (%)

Added

Table 3: Please correct the brackets in the left column

The brackets have been corrected for the age class

Table 3: Specialisms? Maybe you mean specialists

Specialisms has been modified by Provider Specialty

Table 3: Add GPs to the footnotes with a definition

Done

Figure 3: Something is wrong with the legend (grey is not defined) and the numbers on the right side of the figure. Probably some information is lost when fitting on the page?

Indeed, I think this is due to the formatting of the graphic in the text. The graphic has a tendency to move in the Word document this mask some part of the figure. It is well arranged in the pdf version. However, it would be corrected in the final version.

Materials and methods: Delete the word count

Word count has been remove

Was there an ethics approval or waiver for this study? If yes, it should be stated in the Methods section and at the end of the manuscript before the references

Ethics approval section has been added like this

This study did not involve the human person. Ethical approval was granted by the French Data Protection Agency (CNIL) under two separate decrees dated 20 June 2005 and 1 December 2011. These decrees stipulate that access to anonymized health insurance data is granted only for accredited analysts (1), including the ones from Santé publique France, the French national public health agency, to fulfil its public interest missions. Accordingly, the opinion of an ethics committee was not required

Supplementary material: Please define all abbreviations in the footnotes of the tables

Done

A conclusion section summarizing the main findings could be added after the methods section

A conclusion section has been added to the manuscript

The authors could add a small comment regarding how their data could aid in the reduction of unnecessary antimicrobial prescribing, also suggesting methods to implement antimicrobial stewardship practices in the community setting

This work is part of a more global set of actions that includes work carried out by other teams on the proper use of antibiotics. This first step has identified the patients and prescribers associated with antibiotic prescriptions. Thus, the covid-19 pandemic shows us that progress could be made in reducing the prescription of antibiotics in the outpatient sector, in particular those associated with potential inappropriate prescriptions. However, additional studies are needed to better understand drivers of such a reduction and sustained antimicrobial stewardship actions, such as the ones planned in the new French national strategy for preventing antimicrobial resistance (2), will be needed in order for this decrease in antimicrobial prescriptions to be sustainable.

Reviewer 2 Report

The abstract is not clear. The authors did not detail the aim/objectives and methods clearly. 

The authors should provide more details on the number of consultations (per year) with an antibiotic prescription or without. 

Describe how long-term illnesses are defined. 

Table 2 provides the details of antibiotic prescriptions in 2019. What about 2020?

Although the number of prescriptions is less in 2020 due to the COVID-19 pandemic, there is no percentage difference in antibiotic use in France (see table 3). How do authors see this? 

One of the most surprising and strange is that antibiotics use is important to treat several infections, without the details of infections, duration of therapy and appropriateness of its use. This research provides little interest to draw a conclusion. 

In general, antibiotics are widely used in several conditions and as an empirical therapy to treat some conditions. These antibiotics (both narrow and broad-spectrum) were used by the different health professionals following national prescription guidelines or recommendations from the societies. In light of this research, it was not clear how the authors correlate the appropriateness of antibiotic use changed over time. Did COVID-19 give any opportunity to improve the quality of antimicrobial use in France remains unclear? 

The data presented is only descriptive, it's difficult to draw conclusions from a descriptive data. 

Author Response

Dear Reviewer/Editor,

We would like to thank the reviewers and the editor for their thoughtful comments and efforts towards improving our manuscript. All changes have been highlighted in colour in the manuscript.

The abstract is not clear. The authors did not detail the aim/objectives and methods clearly?

The abstract has been modified with detail on the objective and methods of the study. These details have increased slightly the size of the abstract from 281 to 320.

The authors should provide more details on the number of consultations (per year) with an antibiotic prescription or without

Indeed, we are aware that the absence of consultation data for 2020 does not allow us to compare HPCs activities by specialty and between the two years. However, due to the homogeneous renewal of the HPCs demography, we did not provide the same as table 2 for 2020.  We have now added table S4 in the appendix, which provided the characteristics and activity of HPCs in 2020.  As expected, prescribers are fully comparable between 2019 and 2020 on all characteristics but without surprise, with less activities.

Since the average number of prescriptions per consultation is equal to 1.02 in our datasets, 13.4% of consultations resulted in an antibiotic prescription in 2019 against 12% in 2020.

Describe how long-term illnesses are defined?

Long-term illnesses (LTI) are serious and/or chronic illnesses defined by a list of conditions established by decree and which entitle the insured to exemption from co-payment (100% reimbursement) for care related to these illnesses. Examples include cancer, diabetes, long-term psychiatric illnesses, coronary disease, etc.  A patient is classified as suffering from a long-term illness when he or she benefits from the exemption system for long-term illnesses, i.e. when at least one of his or her benefits is fully reimbursed as an exemption from co-payment for a long-term illness during the study period. Thus, the number of cumulative long-term conditions per patient is estimated and classified as follows: 0, 1 and ≥ 2.

The description of this variable has been added to the relevant section in the appendix

Table 2 provides the details of antibiotic prescriptions in 2019. What about 2020?

Indeed, Table 2 shows the overall number of antibiotic prescriptions dispensed by prescribers and per specialty in 2019. Details of the number of prescriptions per specialty are available in Table 3 of the manuscript. However, you will also see details of prescriptions but also of prescriber characteristics in the table S4 that has been added to the supplementary materials.

Although the number of prescriptions is less in 2020 due to the COVID-19 pandemic, there is no percentage difference in antibiotic use in France (see table 3). How do authors see this?

Due to the high statistical power related to the large number of subjects, as seen in the literature (1) and mentioned in statistical analyses section (line 327), we did not use statistical tests to compare the distribution of antibiotic prescriptions between 2019 and 2020. Very marginal differences were statistically significant and a multivariate analysis did not have provide additional information. As indicated in line 105 of the manuscript, the trends observed in the descriptive analysis persisted in the multivariate analysis.

Although the distribution of prescriptions by therapeutic classes may seem comparable between 2019 vs 2020, there are still notable differences. In 2020, the percentage of antibiotic prescriptions was lower for <15 years (18.9 vs 14.9), for Penicillins (52.8 vs 49.6), broad-spectrum penicillin (39.6 vs 35.5), other beta-lactams (10.1 vs 8.5). Also by provider specialty, we observe less antibiotic prescription by GPs (78.0 vs 74.6) but more by dentists (12.2 vs 15.1) in 2020. On the other hand, there are less obvious differences in the distribution of prescriptions according to other factors (long-term illness, age of prescriber, years of experience, etc.).

If we properly understood your request, the overall decrease in antibiotic prescribing observed in 2020 is initially attributable to the impact of restrictive measures that notably limited access to the health system during the pandemic. However, the heterogeneous decrease in prescriptions according to certain characteristics, including patient age and therapeutic classes, could be explained by the absence of circulation of respiratory viruses during the pandemic.

One of the most surprising and strange is that antibiotics use is important to treat several infections, without the details of infections, duration of therapy and appropriateness of its use. This research provides little interest to draw a conclusion.

Indeed, it would have been interesting to have all these details on prescriptions but the data from the National Health Data System do not allow us to establish the link between antibiotic prescriptions and the indication or duration of treatment. As mentioned in the limitations of the study (P12 Lines 252-254), these data do not allow us to discuss the clinical relevance of the prescriptions or whether they are appropriate or not. However, although our study is descriptive, we think it remains important as a first step in identifying the priority targets (patients/prescribers) to be explored (through studies on good use) and towards whom awareness-raising efforts should be directed. Such additional studies are planned within the new national strategy for preventing antimicrobial resistance, which has been quoted in the revised conclusion of the paper.

In general, antibiotics are widely used in several conditions and as an empirical therapy to treat some conditions. The different health professionals following national prescription guidelines or recommendations from the societies used these antibiotics (both narrow and broad-spectrum). In light of this research, it was not clear how the authors correlate the appropriateness of antibiotic use changed over time. Did COVID-19 give any opportunity to improve the quality of antimicrobial use in France remains unclear?

France is one of the biggest consumers of antibiotics in ambulatory care with a high rate of prescription compared to other European countries (2). Given the data available, we were not able to determine the appropriateness of prescriptions issued by health professionals, but we were able to identify those associated with a high prescription rate. In 2020, a decrease in the rate of antibiotic prescriptions is observe but also in the number of consultations. However, there is no indication that the covid-19 pandemic has improved the prescribing habits of health professionals. In fact, the control measures of the pandemic have been successful in controlling the spread of the sars-cov2 virus but have also limited the circulation of respiratory viruses in particular. Given that approximately 50% of consultations for upper respiratory infections (influenza, bronchiolitis, etc.) result in a prescription of antibiotics (3), the reduction in consultations for the latter actually limits opportunities for unnecessary prescriptions. Therefore, and as suggested in the literature (4,5),  we believe that the prescription rate observed in 2020 is probably less inappropriate than what would have been observed if there was an active circulation of winter respiratory viruses. Finally, we believe that the covid-19 pandemic led to more appropriate prescriptions from healthcare professionals, but mostly due to the decrease in respiratory viral infections and without changing their overall prescribing habits. Without targeted actions directed at these health professionals, prescription rates will return to the same level as in the pre-pandemic period.

The data presented is only descriptive; it's difficult to draw conclusions from a descriptive data

As explained in the previous answer, this study, although descriptive, remains important and is part of a more global set of actions that includes work carried out by other teams on the proper use of antibiotics. This first step nevertheless allows us to identify the priority targets (patients/prescribers) to be explored (through good use studies) and towards which awareness-raising efforts should be directed.

References

  1. Demidenko E. The p-Value You Can’t Buy. Am Stat. 2 janv 2016;70(1):33‑8.
  2. Antimicrobial consumption - Annual Epidemiological Report for 2019 [Internet]. European Centre for Disease Prevention and Control. 2020 [cité 29 avr 2022]. Disponible sur: https://www.ecdc.europa.eu/en/publications-data/surveillance-antimicrobial-consumption-europe-2019
  3. Havers FP, Hicks LA, Chung JR, Gaglani M, Murthy K, Zimmerman RK, et al. Outpatient Antibiotic Prescribing for Acute Respiratory Infections During Influenza Seasons. JAMA Netw Open. 8 juin 2018;1(2):e180243.
  4. Kitano T, Brown KA, Daneman N, MacFadden DR, Langford BJ, Leung V, et al. The Impact of COVID-19 on Outpatient Antibiotic Prescriptions in Ontario, Canada; An Interrupted Time Series Analysis. Open Forum Infect Dis. 1 nov 2021;8(11):ofab533.
  5. Gillies MB, Burgner DP, Ivancic L, Nassar N, Miller JE, Sullivan SG, et al. Changes in antibiotic prescribing following COVID-19 restrictions: Lessons for post-pandemic antibiotic stewardship. Br J Clin Pharmacol. 2022;88(3):1143‑51.

Round 2

Reviewer 2 Report

I have no other comments related to the work. The author's should add several limitations related to this research work. 

Author Response

Dear reviewer,
I would like to thank you for your feedback. The limitations related to this work have been presented and discussed in the appropriate section of the manuscript (line 249 to 265). 

Also, as suggested by the editor, the results section has been slightly shortened from 988 to 948 words. 

Thank you for your review

Yours sincerely,